# The Kinetic and Analytical Aspects of Enzyme Competitive Inhibition: Sensing of Tyrosinase Inhibitors

**DOI:** 10.3390/bios11090322

**Published:** 2021-09-08

**Authors:** Raouia Attaallah, Aziz Amine

**Affiliations:** Laboratory of Process Engineering & Environment, Faculty of Sciences and Techniques, Hassan II University of Casablanca, PA 146, Mohammedia 20800, Morocco; raouia.attaallah-etu@etu.univh2c.ma

**Keywords:** competitive inhibition, tyrosinase, progress curve, half-time reaction, graphical plot, amperometric biosensor

## Abstract

An amperometric biosensor based on tyrosinase, immobilized onto a carbon black paste electrode using glutaraldehyde and BSA was constructed to detect competitive inhibitors. Three inhibitors were used in this study: benzoic acid, sodium azide, and kojic acid, and the obtained values for fifty percent of inhibition (IC_50_) were 119 µM, 1480 µM, and 30 µM, respectively. The type of inhibition can also be determined from the curve of the degree of inhibition by considering the shift of the inhibition curves. Amperometric experiments were performed with a biosensor polarized at the potential −0.15 V vs. Ag/AgCl and using 0.1 M phosphate buffer (pH 6.8) as an electrolyte. Under optimized conditions, the proposed biosensor showed a linear amperometric response toward catechol detection from 0.5 µM to 38 µM with a detection limit of 0.35 µM (S/N = 3), and its sensitivity was 66.5 mA M^−1^ cm^−2^. Moreover, the biosensor exhibited a good storage stability. Conversely, a novel graphical plot for the determination of reversible competitive inhibition was represented for free tyrosinase. The graph consisted of plotting the half-time reaction (t_1/2_) as a function of the inhibitor concentration at various substrate concentrations. This innovative method relevance was demonstrated in the case of kojic acid using a colorimetric bioassay relying on tyrosinase inhibition. The results showed that the t_1/2_ provides an extended linear range of tyrosinase inhibitors.

## 1. Introduction

Over the last decade, electrochemical (bio)sensing systems have attracted much attention as a simple and effective alternative to traditional approaches for (bio)analytical studies [1,2,3,4]. The enzymatic biosensors were widely used because of their ability to detect a target substance/analyte with a high specificity through enzyme-catalyzed reactions [5,6,7]. The detection of phenolic [8,9] and other compounds that can be used as a food preservative, and can be hazardous to human health at levels greater than authorized safety standards, such as benzoic acid, sodium azide and kojic acid [10], is of great interest for both quality assurance and consumer protection [11]. Tyrosinase-based amperometric biosensors have been extensively reported in the literature for the detection of various compounds (particularly mono- and di-phenols) due to their rapid response, low cost and low energy consumption [12,13,14]. Tyrosinase catalyzes the oxidation of phenolic compounds to the corresponding *o*-quinones in two stages in the presence of oxygen, which are then further reduced by the electrode, reforming the original phenol, establishing a bio-electrocatalytic amplification cycle [15]. The immobilization and stability of tyrosinase is one of the most important aspects in the potential success of enzyme-based biosensors [16,17]. To date, the carbon black paste electrode has been positioned as an attractive candidate for enzyme immobilization due to the high surface area of carbon black (CB) and long-term stability of enzyme electrodes in various applications of an electrochemical biosensor [18].

Over the last decade, enzyme inhibition has been substantially investigated in pharmacology, toxicology, and analytical chemistry [19]. Competitive inhibitors may be considered as structural analogs of the substrate, and therefore compete for the same active binding sites on the enzyme. Competitive inhibition studies provide information on specific enzyme–substrate complexes and the interactions of specific groups at the active sites. Consequently, pharmaceutical companies synthesize drugs that competitively inhibit the metabolic processes of specific cancer cells and bacteria. Many drugs are competitive inhibitors of specific enzymes [20,21,22]. A typical example of competitive inhibition is the effect of the neuraminidase inhibitor (Relenza drug) on treating the influenza virus [23]. Sulfanilamide, a sulfur-based drug, is an antibacterial agent, since it competitively inhibits the enzyme-catalyzed reaction using *p*-aminobenzoic acid to synthesize folic acid [24]. Such inhibition depends on the inhibitor amount fixed in the active site and is, therefore, related to the inhibitor concentration.

Since the inhibitor binds reversibly, the substrate can compete with it at high concentrations. Thus, a competitive inhibitor does not affect the maximum activity (Vmax) of an enzyme. Besides this, competitive inhibitors increase the Michaelis–Menten constant (Km) of an enzyme because higher substrate concentrations are required to achieve semi-maximal activity [25].

Several graphical plots are currently applied to diagnose inhibition types, such as Dixon, Cornish–Bowden, and Lineweaver–Burk. However, none of these plots alone give satisfactory results [26]. An additional parameter that is important to describe the extent of inhibition is IC_50_ (the concentration of inhibitor that causes 50% inhibition), which is generally used to investigate the impact of drugs on enzymes [27]. By comparing the IC_50_ values measured in different laboratories for the same substrate and the same enzyme, the potency of inhibitor compounds can be assessed.

Some enzymatic kits have been developed for inhibitor analysis [28]. Nevertheless, they are characterized by a narrow linear range because the response curve of the enzymatic inhibition tends to flatten at an inhibitor concentration greater than the inhibition constant (Ki), according to the enzyme kinetic theory [29]. The enzyme generates a product at an approximately linear initial rate for a brief period after the reaction starts. While the reaction continues and the substrate is consumed, the rate slows down continuously (as long as the substrate is not yet at saturation levels). Enzyme tests are usually performed to establish the initial (and maximum) rate when the reaction has progressed only a few percent.

The majority of enzyme kinetic studies focus on the linear part of the enzymatic reactions; although, the percent of substrate conversion is low, and hence the variation in the signal due to the enzymatic product is not sufficient for precise measurements. However, it is also possible to obtain the complete reaction curve measurements and fit these data to a non-linear rate equation. This way of measuring enzymatic reactions is known as a progress curve (i.e., a graphical representation of an enzyme-catalyzed reaction in which the product or substrate concentration is plotted as a function of time) [30]. This approach is suitable for use as an alternative to fast kinetics when the initial rate is too fast to be measured accurately. However, the linear range is always narrow, due to the steady-state appearance at the value of inhibitor concentration higher than the inhibitor constant (Ki) [31].

In the first part of this work, an electrochemical immobilized tyrosinase based biosensor was employed for the quantitative determination of three competitive inhibitors using the degree of inhibition and IC_50_. In the second part, an extended linear range of kojic acid, as a tyrosinase competitive inhibitor, was obtained using a colorimetric method. Herein, we proposed, for the first time, a novel graphical plot for the determination of competitive inhibition through a “half-time reaction” (t_1/2_) estimated by the progress curve.

## 2. Materials and Methods

All chemicals were of an analytical grade and were used without further purification. A phosphate buffer solution (0.1 M, pH 6.8), prepared from sodium phosphate dibasic and sodium dihydrogen phosphate, was used as the supporting electrolyte for all measurements. Tyrosinase (EC 1.14.18.1, from mushroom), catechol, L-tyrosine, Bovine Serum Albumin (BSA), Glutaraldehyde (GA), benzoic acid and sodium azide (NaN_3_) were purchased from Sigma-Aldrich (Burlington, MA, USA), whereas kojic acid was purchased from Alfa-Aesar (Tewksbury, MA, USA). The carbon black paste electrode was prepared by mixing 50% (*w*/*w*) carbon black powder N220 (Ravenna, Italy) with 50% (*w*/*w*) mineral oil (from Fluka, Buchs, Switzerland).

All electrochemical measurements were carried out using a PalmSens Potentiostat electrochemical analyzer provided by PalmSens BV (Utrecht, The Netherlands), controlled by PSTrace 4.0 software. A conventional three-electrode system in an electrochemical cell of 5 mL volume, containing the carbon black paste electrode (CBPE) as a working electrode, a platinum electrode as a counter electrode, and an Ag/AgCl electrode as a reference electrode.

An automated microplate reader (Biotek) was used to measure the absorbance of free enzyme at 490 nm, and the data were evaluated with Gen5 software. OriginPro8 was used as data analysis software.

### 2.1. Preparation of Carbon Paste Electrodes

The ratio between carbon black powder and mineral oil as a binder (1:1 *w*/*w*) was selected following the literature [32,33]. The two compounds were blended carefully in a mortar using a pestle to obtain a homogeneous paste. The final paste was used to fill the cavity (3 mm diameter, 1 mm depth) of the Teflon tube, which served as the sensor body. Then, these carbon black paste electrodes were polished on clean weighing paper to get a smooth surface.

### 2.2. Tyrosinase Immobilization

The immobilization of tyrosinase onto the surface of the carbon black paste electrode (CBPE) was achieved by cross-linking with glutaraldehyde and bovine serum albumin (BSA) as reported elsewhere for several enzyme biosensors [26,33]. In brief, the enzyme solution was first prepared by mixing 15 µL of Tyr (35 U/mL), 7.5 µL of BSA (1% *w*/*v*), and 7.5 µL of glutaraldehyde (0.25% *w*/*v*) on a glass slide. Then, 7.5 µL of the mixture was spread on the surface of a CBPE, and lastly, it was dried at room temperature for 1 h. Before electrochemical measurements, the enzyme electrode was placed under stirring for 10 min in PBS buffer to remove the enzyme not firmly immobilized. The electrodes were maintained at 4 °C in phosphate buffer, pH 6.8 until use.

### 2.3. Enzymatic Assays 

#### 2.3.1. Electrochemical Assays

The tyrosinase was immobilized on the surface of the carbon black paste electrode. The electrochemical measurements of *o*-quinone produced by the tyrosinase-based biosensor, in presence of catechol as substrate, were monitored by an amperometric measurement.

The enzyme activity assays were performed in an electrochemical cell containing 5 mL of 0.1 M phosphate buffer, pH 6.8 at 25 °C. The applied potential was fixed at −0.15 V.

#### 2.3.2. Spectrophotometric Assays

In the case of free tyrosinase, its activity was monitored in the presence of different kojic acid concentrations, by measuring the absorbance values of the enzymatic product at 490 nm using a spectrophotometer. In detail, 50 µL of phosphate buffer solution (0.1 M, pH 6.8), 10 µL of mushroom tyrosinase (18 U/mL), and 20 µL of different kojic acid concentrations were placed in the wells of a 96-well-standard-microplate and mixed with 20 µL of 0.5 mM, 1 mM, 2 mM, and 3 mM L-tyrosine. Substrate conversion was evaluated by measuring the increase in absorbance at 490 nm. The absorbance was sampled every 1 min, allowing each progression curve to be constructed in 40 min.

### 2.4. Biosensor Response Measurements

To perform each amperometric measurement, the CBPE-Tyr electrode was dipped into an electrochemical cell containing 5 mL of 0.1 M phosphate buffer solution (pH 6.8) under constant magnetic stirring. The applied potential was fixed to −0.15 V versus Ag/AgCl. Once the baseline was established (10 min approximately), a defined amount of catechol solution was added to the measuring cell. A large reduction current was observed due to the generation of *o*-quinone, and a plateau corresponding to the steady-state response was reached in approximately 20 s. Then, the inhibitor solutions were added consecutively each time a plateau was reached (approximately every 30 s). The addition of tyrosinase inhibitors causes an inhibition of the enzyme, consequently decreasing the amount of liberated enzymatic reaction products.

### 2.5. Competitive Inhibition of Tyrosinase

Under competitive inhibition, the inhibitor competes with the substrate to bind to an active site. When the inhibitor binds to the active site, an enzyme–inhibitor (EI) complex is formed, and the enzyme cannot react until the inhibitor dissociates (Figure 1).

To evaluate the degree of inhibition (*I*%) of the tyrosinase activity, signal responses in the absence (*I*_0_) and the presence of an inhibitor (*I*_1_) were measured with either an electrochemical or colorimetric method. The degree of inhibition (*I*%) is expressed as follows: (1)$$I\left(\%\right)=\frac{I_{0}-I_{1}}{I_{0}}\times 100\;$$

When using the amperometric biosensor, the CBPE/Tyr modified electrode was immersed into a phosphate buffer solution at pH 6.8, under stirring and a fixed potential of −0.15 V. After stabilizing the baseline current, a fixed amount of catechol (substrate) was added to reach a steady-state current (*I*_0_) before adding the inhibitor. A large decrease in the current was observed due to the addition of catechol. The concentration of inhibitors was increased to inhibit tyrosinase activity, and the current decrease (*I*_1_) was recorded [26]. Appendix A illustrated the typical behavior of the biosensor response in the absence (*I*_0_), and presence of an inhibitor (*I*_1_).

All experiments were repeated at least thrice. The data presented here are from representative experiments. The mean values ± SD are reported.

## 3. Results and Discussion

The use of an amperometric enzyme biosensor based on immobilized tyrosinase to determine the concentration of the inhibitor that causes 50% inhibition (IC_50_), is described in the following section. Indeed, the type of inhibition can be checked from the curve of the degree of inhibition versus the inhibitor concentration. To this purpose, different substrate concentrations were tested in order to determine their effect on the IC_50_ variation, and the inhibition type can be directly detected. Furthermore, free tyrosinase was used to develop a new and straightforward graphical plot to identify the inhibition type using the progress curve and to determine kojic acid at an extended linearity.

### 3.1. Analytical Performances of Tyrosinase Biosensors for Substrate Determination

The cyclic voltammograms obtained using a CB-Tyr modified carbon paste electrode before and after the addition of 0.5 mM catechol in 0.1 M phosphate buffer solution (pH 6.8) are shown in Appendix A. The potential was scanned between −1 and 1 V versus Ag/AgCl at a scan rate of 50 mV s^−1^. Upon the addition of catechol, an expected oxidation peak occurred around +0.3 V in the anodic potential scan was attributed to the oxidation of the catechol to *o*-quinone, and a reduction peak corresponded to the reduction in the *o*-quinone, which appears at around −0.15 V.

The developed tyrosinase biosensor catalyzes the oxidation of catechol to *o*-quinone that can be electrochemically reduced for the amperometric detection of phenol [15]. The following equation can summarize the biocatalytic process for the oxidation of catechol in the presence of tyrosinase:$$Catechol+\frac{1}{2}O_{2}\overset{Tyrosinase}{\to }O-quinone+H_{2}O$$

Figure 1 demonstrates the linear calibration curve of the tyrosinase–CBPE biosensor with successive addition of catechol concentrations to the 0.1 M phosphate buffer solution (pH 6.8). The inset of Figure 1 displayed a typical amperogram recorded at the tyrosinase-CBPE biosensor at −0.15 V. The response time was within 8–10 s, which indicated a rapid process. A linear relationship between the current and catechol concentration in the range of 0.5–38 μM was achieved, with a correlation coefficient of 0.982. The biosensor displayed a high sensitivity of 66.55 mA M^−1^ cm^−2^ to catechol, and a limit of detection of 0.35 μM (signal/noise [S/N] = 3).

The obtained analytical characteristics of CBPE-Tyr for the determination of catechol were compared with the most relevant previous reports. According to Table 1, a relatively high sensitivity was obtained. Furthermore, the detection limit for catechol obtained in the present work was lower than most of the other tyrosinase-modified electrodes due to the high surface area of CB and to the high efficiency of the bio-electrocatalytic amplification cycle, which is due to the close proximity between CB nanoparticles and enzymes.

To evaluate the stability of the proposed biosensor, the enzyme-modified electrodes were stored in a 0.1 M phosphate buffer solution (pH 6.8) at 4 °C, and periodically interrogated. Indeed, the biosensor response in the presence of 20 µM catechol solution was measured and recorded as a function of time. The developed biosensor revealed a high stability, as shown in Appendix A. The biosensor retained up to 95% and 60% of its initial response after 10 and 30 days of storage, in 0.1 M phosphate buffer (pH 6.8) at 4 °C, respectively.

### 3.2. Analytical Performances of Biosensors Based on Immobilized Tyrosinase for Inhibitor Determination

The biosensor developed was explicitly conceived to detect competitive tyrosinase inhibitors, such as benzoic acid, sodium azide, and kojic acid by testing their potency. Many factors affect the final results, including the amount of immobilized enzyme, method of immobilization, substrate type, substrate concentration, contact time between enzyme and substrate and between enzyme and inhibitor, pH, temperature, applied potential, and stirring rate of the solution [44]. To properly compare the inhibition of the studied compounds, we worked under the same experimental conditions as described above.

It was verified that the CBPE (i.e., without tyrosinase) did not respond to the above inhibitors under the experimental conditions. Indeed, kojic acid, benzoic acid, and sodium azide were found to give no amperometric signal. Similarly, blank experiments were also performed on the CBPE/Tyr without the addition of the catechol substrate. No amperometric response was seen for any of the tested inhibitors.

The biosensor response to catechol was investigated in the presence of sodium azide, benzoic acid, and kojic acid as inhibitors of the enzymatic reaction reducing the quinone formed. It can be expected that, with higher concentrations of the inhibitor under study, the enzymatic reaction will occur at a low rate, and, consequently, the current supplied by the biosensor will gradually decrease.

Two concentrations of the catechol substrate (20 and 200 µM) were tested to investigate the effect on the inhibition in the presence of various concentrations of inhibitors. In the case of competitive inhibition, there was no inhibition in the presence of a high concentration of the substrate, since the substrate competes with the inhibitor for the active site of the enzyme [28].

In this regard, Figure 2 demonstrated the calibration curve of kojic acid constructed by plotting the degree of inhibition as a function of kojic acid concentration in the presence of two catechol concentrations. As observed, the inhibition percent of kojic acid increases with the increase in substrate concentration. Indeed, by increasing the substrate concentration from 20 µM to 200 µM, the inhibition curve shifted to the right and the IC_50_ increased from 30 µM to 58 µM, demonstrating that tyrosinase inhibition by kojic acid is a competitive inhibition. This was also confirmed by other previous work in the literature [10,45]. We previously reported [27] that the degree of inhibition is equal to [I]/([I] + [IC_50_]), which means a hyperbolic relationship exists between the degree of inhibition and concentration of an inhibitor. This means that at a high concentration of inhibitor (e.g., from 20 μM to 40 μM vs. from 60 μM to 80 μM kojic acid), the degree of inhibition tends to deviate from linearity.

Further experiments of tyrosinase inhibition were carried out for benzoic acid and sodium azide. The IC_50_ values for all tested inhibitors against tyrosinase were summarized in Table 2. As seen in Table 2, the IC_50_ value increases with increasing substrate concentration. Given this, the use of a high concentration of substrate will not produce significant inhibition responses.

The inset of Figure 3 showed a typical time-dependent response obtained for the tyrosinase CBPE biosensor by catechol injection followed by the successive addition of inhibitors. It can be seen that a decrease in the current was obtained after the addition of 20 µM catechol. Successive additions of an inhibitor led to a diminution of the reduction current. It indicated that kojic acid, benzoic acid, and sodium azide competed with catechol at the electrode surface.

The inhibition percentage (*I*%) was obtained by varying the concentration of inhibitors at a fixed substrate concentration of 20 µM (Figure 3).

Kojic acid was tested in the concentration range from 2.5 µM to 97.5 µM, which corresponds to the inhibition range of 13–74% of catechol response (Figure 3a). Identical experiments were carried out for benzoic acid in the concentration range from 5 µM to 275 µM, which corresponds to the inhibition range of 8–69% (Figure 3b) and also for sodium azide in the concentration range from 0.01 mM to 1.57 mM, which corresponds to the inhibition range of 7–64% (Figure 3c).

The concentration that inhibited 50% of the catechol signal (IC_50_) was calculated from the degree of inhibition curves by plotting the percent inhibition versus inhibitor concentration. The IC_50_ values were determined to be 119 µM, 1480 µM, and 30 µM for benzoic acid, sodium azide, and kojic acid, respectively.

### 3.3. Kinetic Study and Analytical Aspects of Free Tyrosinase

In this section, free tyrosinase will be challenged with inhibitors and the residual enzyme activity will be measured over time. The kinetic and analytical aspects will be highlighted.

The Michaelis–Menten equation generally describes the kinetics of the enzyme reaction:(2)$$V_{0}=\frac{Vmax.S}{Km+S_{0}}\;$$

*V_max_ =* Maximum velocity.

*V*_0_*=* Velocity in the absence of the inhibitor.

*K_m_* = Michealis constant of the substrated.

*S* = Substrat concentration.

In agreement with the literature [29], Equation (2) can be integrated and rearranged to: (3)$$\frac{S_{0}-S}{t}=-Km.\;\left(\frac{2.3}{t}.log\frac{S_{0}}{S}\right)+Vmax\;$$

The curve of (*S*_0_
*− S*)/*t* versus *1*/*t* 2.3 *log*(*S*_0_/*S*) is used to determine the kinetic parameters *Km* (slope) and *Vmax* (intercept).

The integrated half-time equation is shown as follows: (4)$$t_{1/2}=\left(2.3\;log\;2.\;\frac{Km}{Vmax}\right)+\left(\;\frac{S_{0}}{2.Vmax}\right)$$

However, in the presence of a competitive inhibitor, Equation (5) represents the Michaelis–Menten equation:(5)$$V_{i}=\frac{Vmax.\left[S\right]}{Km\;\left(\;1+\frac{I}{Ki}\right)+\left[S\right]}$$

*Vi* presents the velocity in the inhibitor’s presence, *I* is the concentration of the inhibitor, and *Ki* presents the dissociation constant of the enzyme-inhibitor (EI). 

The integrated Michaelis–Menten equation for competitive inhibition is: (6)$$t’=\frac{Km}{Vmax}.\;\left(1+\frac{I}{Ki}\right).2.3.log\frac{S_{0}}{S}+\frac{S_{0}-S}{Vmax}$$

As mentioned in this integrated rate Equation (6), time (*t*’) is not linearly related to the inhibitor concentration (*I*), since the log conversion rate (*S*_0_/*S*) varies with time. However, the measurement at the time required to achieve 50% of the substrate conversion in the presence of an inhibitor (*t*’_½_) leads to a linear relation with the inhibitor concentration.

The integrated half-time equation in the presence of an inhibitor can be written as follows:(7)$$t{’}_{1/2}=\frac{Km}{Vmax}.\;\left(1+\frac{I}{Ki}\right).2.3log2+\frac{S_{0}}{2.Vmax}$$

The progress curves of the tyrosinase reaction in the absence and presence of kojic acid are shown in Figure 4a. The $t{’}_{1/2}$ calculated from the graph was plotted against the inhibitor concentration, and an extensive linear range was seen with R^2^ = 0.997 (Figure 4a). This approach allowed us to obtain an extended linear range in comparison with the use of conventional initial rate methods for the inhibitor determination, whose upper limit of the linear range is around IC_50_. Indeed, the IC_50_ for kojic acid was equal to 30 µM using an amperometric biosensor. On the other hand, the method suggested in this work revealed a wide linear range up to 300 µM (Figure 4b).

As well as the analytical aspects, the diagnosis of the inhibition type can be ascertained from the plot *t*’_1/2_ versus inhibitor concentrations. The substrate concentration will be varied to determine its effect on the half-time reaction, and after this change, we can directly determine the type of inhibition.

In the case of competitive inhibition:(8)$$t{’}_{1/2}=\;\left[\left(2.3\;log2.\;\frac{Km}{Vmax}\;\right).\left(1+\frac{\left[I\right]}{Ki}\right)\right]+\left(\frac{S_{0}}{2.Vmax}\right)$$
(9)$$t{’}_{1/2}=\left(2.3\;log\;2.\;\frac{Km}{Ki.Vmax}\;\right)\left[I\right]\;+\left(2.3\;log\;2.\;\frac{Km}{Vmax}\;\right)+\left(\frac{S_{0}}{2.Vmax}\right)$$
(10)$$t{’}_{1/2}=\left(2.3\;log\;2.\;\frac{Km}{Ki.Vmax}\;\right)\left[I\right]+A$$

It should be noted that without an inhibitor:(11)$$t_{1/2}\;=\;(2.3\;log\;2.\;\frac{Km}{Vmax}\;)\;+\;\left(\frac{S_{0}}{2.Vmax}\right)=A$$

According to Equations (9) and (10), any change of the substrate concentration (*S*_0_) will affect the value of the intercept (*A*), but the value of the slope (2.3 *log* 2 × $\frac{Km}{\;K_{i}.Vmax}$) remains unvaried. To demonstrate the applicability of this new approach, we measured *t*’_1/2_, the time required to reach the 50% of L-tyrosine conversion in the presence of different concentrations of kojic acid using tyrosinase as a free enzyme and tyrosine as a substrate at concentrations varying from 0.5 to 3 mM. In other words, the experiment of Figure 4 was repeated at a concentration of tyrosine equal to 0.5, 1, and 3 mM. As indicated in Figure 5, increasing the substrate concentration causes parallel lines with a high intercept and unvaried slope. These results confirm the competitive inhibition of tyrosinase by kojic acid and are in agreement with the previous literature [10,45] and with the above results, obtained by measuring the degree of inhibition using an amperometric tyrosinase biosensor.

## 4. Conclusions

Herein, we describe for the first time, an easy-to-use graphical plot for the determination of a reversible competitive inhibition type. Thus, the use of a progress curve for inhibitor determination allows for a plot of the time required to achieve 50% of a substrate conversion versus inhibitor concentrations. It is important to note that using the integrated progression curve equation allows for an extended linear range, which is highly required for inhibitor determination in undiluted samples. The proposed graph was successfully applied for tyrosinase using L-tyrosine as a substrate and kojic acid as an inhibitor. Furthermore, the diagnosis of competitive inhibition can be performed by plotting the degree of inhibition versus inhibitor concentration. Amperometric tyrosinase biosensor was successfully employed using the degree of inhibition to determine benzoic acid, sodium azide, and kojic acid at the micromolar level.

## Data Availability

The study did not report any data.

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
