# Peer review of "The Kinetic and Analytical Aspects of Enzyme Competitive Inhibition: Sensing of Tyrosinase Inhibitors"

_biosensors, 2021, doi:10.3390/bios11090322_

Round 1

Reviewer 1 Report

The manuscript by Attaallah and A. Amine: Kinetic and analytical aspects of enzyme competitive inhibition: Sensing of Tyrosinase Inhibitors” refers to an amperometric biosensor based on tyrosinase immobilized onto a carbon black paste electrode to detect competitive inhibitors. The subject area of the manuscript is quite interesting, and it would certainly add a scientific contribution to the relevant field. I recommend the publication in ‘‘Biosensors’’ after revision. The following suggestions are provided for the authors' revising manuscript.

  1. It’s better to include analytical performance like LOD, linear range, sensitivity in abstract because these are necessary.
  2. The authors need to elaborate in detail the mechanism of
  3. In Fig. 3, It would be knowledge lavishing if the author can explain why current is higher in first spike but lower in other spikes of catechol ??
  4. These are some closely related articles on EC sensors which authors need to read and cite to improve this manuscript: Journal of the Electrochemical Society 2019, 359-369‏; ACS Appl. Mater. Interfaces 2018, 10, 36675−36685; Biosensors and Bioelectronics 97 (2017) 352–359; ACS Appl. Mater. Interfaces 2021, 13, 5, 6023–6033; Advances in Colloid and Interface Science 262 (2018) 21–38; Anal. Chem. 2019, 91, 3912−3920; Sensors and Actuators B 239 (2017) 243–252.
  5. Insets of Fig.3 are not visible.
  6. Nothing has been said about fouling which may affect the sensing performance.
  7. The modification of carbon black paste electrode may change the peak current value so it is recommended to use current density instead of using just current. It is suggested to use current density in all electrochemical measurements following and citing these articles Microchimica Acta (2019) 186:61 and Analytica Chimica Acta 1047 (2019) 197-207.
  8. Authors are advised to include a comparison table with at-least 10 articles to highlight their work.

Reviewer 2 Report

In the work by Attaallah and Amine, the authors characterize two strategies for the determination of the inhibition capabilities of tyrosinase inhibitors (benzoic acid, sodium azide and kojic acid). The work is well written, but I do not understand the innovation about this work.

Moreover, data in Table 1 and in Figure 4 and Figure 5 are reported without the associated error bars and the authors don’t report on how many experiments they calculate the averaged values.

I suggest to add a section in which the authors explain why the current (signal of the amperometric detection) decreases when tyrosinase interacts with the substrate (catechol) and increases in presence of inhibitor. It will be useful for the readers.

The authors should compare their result with other techniques and describe why they have selected those molecules as inhibitor (e.g. they are present in bodily fluids etc.)

30% of self-citations is a bit too high.

I think that this work is more suitable for a lower impact journal.

To be published, the draft must be improved.

Reviewer 3 Report

The overall idea of the paper is worth considering. But, major revision is needed to be done to answer some unclear parts of the context. Please find the following details as feedback.

General comment:

  1. The conceptual discussion should be addressed in both biosensors and analyzing the inhibition mechanism.
  2. Some characterization tests for the fabricated biosensors should be done.
  3. The context has a lack of constancy in the captioning. (e.g., Fig. 1S in line 139 and Figure S2 in line 174).
  4. The abstract could be written in a more elaborative and informative way.
  5. The introduction part should be completed and some more relevant papers in this regard should be added. 13 references for the introduction and elaborating a concept is not enough.

Detailed comments:

  1. In the biosensor fabrication and immobilizing the enzyme on it, the bonding characterization should be done.
  2. The condition of the storage for the durability tests in lines 174 and 175 should be mentioned.
  3. To confirm the O-quinone reduction during the biosensing, the cyclic voltammetry test should be done. The voltage in line 164 should be defined by CV tests.
  4. To define the linear range of the biosensor, more data is needed from 20 to 80 μM. It seems to have a deviation in linearity in concentrations before 80 μM.
  5. Why have the inhibition tests been done in 200 μM catechol? 200 μM catechol is completely out of the range of your calibration curve. The calibration curve should be tested in order to see if it is still linear in higher concentrations or not. Therefore, the inhibition percentage is not accurate.
  6. It is better to elaborate on the mechanism of competitive inhibition. Does this mechanism linearly depend on the concentration of the inhibitor? If not, what is the reason behind the lower inhibition effect by the addition of same concentration (e.g., from 20 μM to 40 μM vs from 60 μM to 80 μM Kojic Acid).
  7. In figure 3, (a) and (b) are more similar to each other and (c) differs in behaviour. Why could this be happening and what is the reason behind this linearity in higher concentrations of sodium azide?
  8. The numbers of cited references are very low (only 17). Out of these 17 references, 6 references are self-cited from the second author. More literature should be cited for a full-paper manuscript.

Round 2

Reviewer 2 Report

The work can be published now

Reviewer 3 Report

All comments are addressed. I have no further comments.